# How Aqueous Solvation Impacts the Frequencies and Intensities of Infrared Absorption Bands in Flavin: The Quest for a Suitable Solvent Model

**DOI:** 10.3390/molecules29020520

**Published:** 2024-01-20

**Authors:** D. P. Ngan Le, Gary Hastings, Samer Gozem

**Affiliations:** 1Department of Chemistry, Georgia State University, Atlanta, GA 30303, USA; nle51@student.gsu.edu (D.P.N.L.); ghastings@gsu.edu (G.H.); 2Department of Physics and Astronomy, Georgia State University, Atlanta, GA 30303, USA

**Keywords:** flavin, FMN, FAD, riboflavin, vitamin B2, infrared spectroscopy, FTIR, solvent models, PCM, QM/MM

## Abstract

FTIR spectroscopy accompanied by quantum chemical simulations can reveal important information about molecular structure and intermolecular interactions in the condensed phase. Simulations typically account for the solvent either through cluster quantum mechanical (QM) models, polarizable continuum models (PCM), or hybrid quantum mechanical/molecular mechanical (QM/MM) models. Recently, we studied the effect of aqueous solvent interactions on the vibrational frequencies of lumiflavin, a minimal flavin model, using cluster QM and PCM models. Those models successfully reproduced the relative frequencies of four prominent stretching modes of flavin’s isoalloxazine ring in the diagnostic 1450–1750 cm^−1^ range but poorly reproduced the relative band intensities. Here, we extend our studies on this system and account for solvation through a series of increasingly sophisticated models. Only by combining elements of QM clusters, QM/MM, and PCM approaches do we obtain an improved agreement with the experiment. The study sheds light more generally on factors that can impact the computed frequencies and intensities of IR bands in solution.

## 1. Introduction

FTIR spectroscopy measurements are typically carried out in the condensed phase, where intermolecular interactions like hydrogen bonding impact the frequencies and broadening of spectral bands [1]. To simulate condensed-phase FTIR spectra, computational models must account for the effect of intermolecular interactions on molecular vibrations. Understanding how to best model FTIR spectra in the condensed phase can be especially useful for developing suitable protocols to simulate FTIR spectra of biomolecules, where shifts in IR frequencies (often measured using difference spectroscopy) encode important information about changes in local interactions or macromolecular structure [2,3,4]. For instance, FTIR spectroscopies have been used to probe intermediates formed following the photoexcitation of several flavin-binding photoreceptors [5,6,7,8,9,10,11,12,13,14,15,16,17]. FTIR spectra of protein-bound flavin have also been simulated using hybrid quantum mechanical/molecular mechanical (QM/MM) methods to accompany such experiments [4,18,19,20,21,22].

Here, we focus on flavin in solution, a benchmark system for which vibrational frequencies have been reported experimentally [7,22,23,24,25,26,27,28,29] and computationally [28,29,30,31,32,33,34]. The accurate simulation of vibrational frequencies and intensities is important not only to reproduce FTIR difference signals, but also to compute Franck–Condon factors for electronic transitions [30,32,35,36,37,38,39,40]. Several methods are available for simulating condensed-phase FTIR spectra, but they typically fall into one of two categories. The first involves running molecular dynamics (MD) simulations and extracting the vibrational frequencies from the Fourier transform of the autocorrelation functions [41,42]. The second approach is the direct calculation of vibrational frequencies and intensities using normal mode analysis. The latter assigns IR bands to specific vibrational modes, which is useful for understanding the effect of local interactions [4,43]. We will use the normal mode analysis approach.

Lumiflavin is a reduced model system that contains the tricyclic isoalloxazine ring common to biomolecules like riboflavin (vitamin B2), flavin mononucleotide (FMN), and flavin adenine dinucleotide (FAD), but where the ribose side chains are replaced by a methyl group (see insets in Figure 1). Recently, we used the polarizable continuum model (PCM) and gas-phase quantum chemical solute–solvent cluster models (QM clusters) to simulate the vibrational spectrum of lumiflavin in solution [34]. These calculations were compared to the FTIR spectrum of FMN in D_2_O in the 1450–1750 cm^−1^ range where prominent C=N and C=O stretching frequencies appear. The goal was to understand the effect of hydrogen bonding on those stretching frequencies and to determine whether simple computational models could reproduce the aqueous-phase FTIR spectra. As in that study, we will focus here on four bands: two carbonyl stretches (C_2_=O and C_4_=O) and two modes previously assigned to coupled in-plane and out-of-plane C=N stretches (C=N_in_ and C=N_out_). It has been shown through local mode analysis that the ribose and phosphate moieties do not contribute to the IR spectrum in this energy range [4], so only the isoalloxazine group in FMN contributes to those stretching frequencies. As noted in our previous study [34] and again towards the end of this manuscript, those modes are mixed, especially in the case of C=N_out_ where C=C modes are coupled, but we will keep the labels for consistency with earlier assignments [14].

While the PCM and QM cluster models successfully reproduce the relative frequencies of the prominent stretching modes, there are a few notable differences between the computed and experimental spectra. Experimentally, the C=N_out_ to C=N_in_ intensity ratio is around 5:1. The QM cluster models incorrectly predict similar intensities for the two peaks, while the PCM model predicts a ratio that is more in line with the experiment (Figure 1). However, the PCM calculations predict a prominent peak near 1530 cm^−1^ that is not observed experimentally [34]. These differences are also reflected in the literature [4,18,19,20,21,22,28,29,30,31,32,33,34], where different models can report different relative frequencies and intensities of flavin’s IR bands.

Here, we revisit the effect of solvation on the double-bond stretching vibrational frequencies and intensities of flavin using QM/MM [44,45,46] simulations of lumiflavin in water, using the ONIOM approach [47,48]. The goal is to compare the solvent models and systematically understand which factors influence the band frequencies and intensities. This work will provide guidelines for suitable computational protocols aimed at simulating FTIR difference spectra of flavin-binding proteins.

## 2. Results and Discussion

Molecular dynamics simulations were carried out for lumiflavin in solution. Snapshots from the MD simulation then served as starting points for B3LYP/6-31+G**/TIP3P QM/MM ONIOM calculations [47]. Vibrational frequencies were computed at the same level of theory after hydrogen atoms were replaced with deuterium for all water molecules and for the exchangeable proton on flavin’s N_3_. This is done for consistency with the experiments, which were also performed in D_2_O to avoid bands from intense water-bending vibrations that can overwhelm the flavin bands of interest [34]. 

It is known that hybrid DFT methods slightly overestimate computed frequencies relative to experiments [49]. To correct for this, the computed spectra were adjusted using a constant scaling factor that was chosen for each model to match the computed ν_C=N(out)_ vibrational frequency with the experimental one (1548 cm^−1^). The scaling factor used for each model is indicated in each of the figures in Section 2. QM/MM vibrational frequencies were computed using several different protocols represented in Figure 2. 

In protocol **M1**, only the lumiflavin is treated at the QM level of theory while all water molecules are treated at the MM level (Figure 2A). We test the effect of using a larger solvent box size such that there is at least 12 Å from any lumiflavin atom to the edge of the box, instead of the 3 Å used by default (protocol **M1-Large**). We also tested a charge equilibration method (QEq) for the water molecules [50], which is a variable-charge model, instead of using the fixed-charge TIP3P model (protocol **M1-QEq**).

In protocol **M2**, we included some of the water molecules in the QM subsystem (Figure 2B). Since carbonyl peaks are the most sensitive to hydrogen bonding, we included water molecules that are close to the carbonyl oxygen atoms. Specifically, any water molecule with an atom that is within 3.5 Å of the lumiflavin carbonyl oxygen atoms was selected using VMD and included in the QM region [51]. Using this criterion, between 5 and 13 water molecules were treated quantum mechanically in each snapshot. These QM waters are optimized along with the lumiflavin during the ONIOM QM/MM optimization step.

In protocol **M3**, we account for the long-range effects of solvation implicitly through the ONIOM-PCM approach [50,52]. We used the ONIOM-PCM/X approximation, where the PCM cavity is constructed around the entire QM/MM system and used for both the low-level and high-level calculations (Figure 2C). QM water molecules are also are optimized along with the lumiflavin, just as in M2.

Initially, QM/MM frequency calculations were performed for only a few select QM/MM snapshots obtained from the MD simulations. Figure 3 shows ten simulated FTIR spectra obtained from different snapshots of an MD simulation of lumiflavin in water. “Stick” spectra were calculated using the B3LYP/6-31 + G* ONIOM method and broadened by convolution with 8 cm^−1^ wide (FWHM) Gaussian functions. There are significant variations between the different snapshots, especially for the carbonyl (C=O) stretching bands that show a strong sensitivity to variations in the water micro-environment. This indicates that a single QM/MM calculation is unlikely to be representative of the average environment. Therefore, the remainder of this work will present and discuss the ensemble result from 100 QM/MM calculations for each protocol.

The sum of 100 QM/MM calculations carried out using protocol **M1** is shown in Figure 4B. The vibrational frequencies coming from different calculations merge into just a few prominent broad bands. The relative frequencies of the C=N bands in the composite spectrum appear to match well with the experimental data, but the relative intensities are not in line with that observed experimentally. In the case of the C=O bands, neither the calculated frequencies nor intensities appear to match well with the experiment.

Increasing the solvent box size such that the lumiflavin is 12 Å from the edge of the box, instead of 3 Å, had a limited effect on the quality of the calculations (compare panels B and C in Figure 4). Although the spectra in Figure 4B,C are obtained from different MD simulations and therefore different QM/MM structures, the two sets of calculations are highly consistent. This indicates that the disagreement between calculated and experimental spectra is due to a deficiency in the computational approach/model rather than being due to insufficient sampling.

We tested using a charge equilibration method (QEq), which estimates the charge on each of water’s oxygen and hydrogen atoms based on their coordinates. We reasoned that a more flexible charge model may be able to better capture the interactions between lumiflavin and nearby water molecules. While this approach did alter the calculated spectra (compare panels B and D in Figure 4), it arguably made the agreement with the experiment worse.

To improve the description of the hydrogen bonding with lumiflavin’s carbonyl groups, we included a few water molecules in the QM subsystem of the ONIOM calculations. The results are shown in Figure 4E. This did result in some improvement in the agreement between the calculated and experimental spectra, but the experimental C_2_=O and C_4_=O bands remain poorly reproduced in the calculation while the C=N stretch band relative intensities still do not match the experiment.

Using a simple QM/MM approach that treats water using point charges did not yield a quantitatively accurate FTIR spectrum for flavin in solution. Even treating water molecules close to the lumiflavin quantum mechanically seems to have only a small effect. Therefore, we decided to consider a hybrid QM cluster/MM/PCM approach using ONIOM-PCM/X (protocol **M3**). The spectrum calculated using this hybrid method is shown in Figure 4F and is a significant improvement compared to the other models used in this work. Protocol **M3** combines favorable features of both the QM cluster and PCM calculations from Figure 1 and captures the relative frequencies and intensities of the four prominent bands more accurately than other protocols tested so far. We note that the broad band labeled as C_2_=O likely contains two underlying vibrational frequencies (a mix of C_2_=O and C=N_out_) that contribute to the non-Gaussian shape of the band [31,34]. This broadening appears to be reproduced well in Figure 4F. Finally, we find that the prominent band that appears in the PCM calculation in Figure 1 at 1530 cm^−1^ is now less prominent in the ONIOM-PCM calculations due to being broadened and partially merged with the C=N_out_ band.

In PCM, the solvent dielectric responds to the presence of the solute with mutual polarization (i.e., the continuum solvent dipoles respond to the electric field of the solute while the solute wave function is updated self-consistently) [53]. Since the PCM cavity is constructed around the entire QM/MM (solute and explicit solvent) system, it introduces a long-range dielectric response. This response has a direct effect on the solute’s wave function but may also affect the solute’s interaction with the QM solvent. Indeed, it appears that the latter may be important; the average O---O distance over 100 snapshots between the closest water molecule to the C_2_=O carbonyl oxygen is 2.85 Å in protocol **M2** but is reduced to 2.77 Å with the introduction of PCM solvation in protocol **M3**. Similarly, for the C_4_=O oxygen, the average O---O distance is 2.97 Å in protocol **M2** and decreases to 2.90 Å in protocol **M3**.

To understand why the PCM or ONIOM/PCM results differ from those obtained in cluster or regular ONIOM calculations, we reoptimized the structure and computed vibrational frequencies using PCM solvation with varying dielectric constant *ε*. We then carried out Potential Energy Distribution (PED) calculations using Vibrational Energy Distribution Analysis (VEDA) version 4.0 [54]. VEDA indicates the extent to which stretching, bending, and torsional motions contribute to specific normal modes.

First, in Figure 5B,C, we show unscaled IR spectra in the gas phase and in PCM water (ε = 78.355), respectively. Without scaling, the vibrational frequencies of all the prominent bands in the 1450–1750 cm^−1^ range downshift upon solvation. We also see that the band near 1580 cm^−1^ (yellow square, previously labeled C=N_out_) becomes significantly more intense in the PCM-calculated spectrum. A nearby band (red x), also near 1580 cm^−1^ and not visible in the gas-phase-calculated spectrum due to a low intensity, appears as a prominent band in the PCM-calculated spectrum. A third band (blue circle), just above 1600 cm^−1^, is also downshifted but has a similar intensity in both the gas-phase- and PCM-calculated spectra. In Figure 5D,E, we plot the frequency and intensity, respectively, of those three bands as a function of varying the solvent dielectric constant. To produce this result, we varied the dielectric constant for water from 2 to 100 while keeping all other PCM parameters the same.

Figure 5F–I show the contribution of C=N_out_, C=N_in_, C=C, and C=N_polar_ bonds to those vibrations, respectively. The molecular groups are shown in magenta in the lumiflavin molecules in the inset. Here, C=N_out_ is an out-of-phase combination of N_1_=C_10a_ and C_4a_=N_5_ stretching, while C=N_in_ is the corresponding in-phase combination. C=C is primarily a C_5a_=C_9a_ stretch, while C=N_polar_ is a combination of stretching modes involving the N_10_ and N_3_ atoms. We find that there is a strong correlation between the band intensities and their vibrational character. Specifically, the strong increase in the intensity of the band labeled with a yellow square in Figure 5B correlates with an increasing contribution of C=N_polar_ bonds to the vibration (Figure 5I). The reason for the associated increase in the intensity of the band labeled with a red x in Figure 5B is not clear, but it may be due to some small contribution (<10%) of the same C=N_polar_ bonds to that normal mode. VEDA 4.0 does not print data on molecular groups that make smaller than a 10% contribution to a normal mode.

In summary, band intensities are related to the mixing of the wave function character of the normal modes (e.g., as in Fermi resonances). This mixing is altered upon solvation, leading to both frequency and intensity changes, which alters the Fermi resonances formed by each band. An increasing dielectric environment mostly downshifts normal modes, but the extent of the downshift is slightly different for each band (Figure 5D). In a lower dielectric environment, normal mode intensities are closer together (Figure 5E). Experimental support for this predicted result may have been obtained from FTIR experiments on several flavin derivatives in KBr disks, where it was found that the two C=N mode intensities were more closely matched [55]. Further FTIR experiments on flavin in different polar solvents (e.g., compare lumiflavin in deuterated water [24], lumiflavin in sucrose [7], and riboflavin in deuterated acetonitrile [56]) also indicate significant intensity variations of the two C=N modes, again supporting our calculated prediction.

Computations by Tavan and co-workers similarly found notable differences in the intensities of the C=N normal modes when comparing lumiflavin in the gas and solvent phases [31]. They were able to reproduce the relative intensities of those bands using QM/MM calculations similar to our M1 protocol, but with a number of variations in the method used. We tested our M1 protocol using the same functional and basis set as they used (BP86/TZVP) but could not reproduce the relative intensities of those bands. Therefore, we attribute the difference in our results to other differences in our protocols.

Here, we looked more closely at the factors that affect the relative intensities of the IR bands in solution. We found that to reproduce the experimental relative intensities, what is needed is a solvent model that correctly reproduces the absolute differences in frequencies of both intense and non-intense bands in the IR spectrum so that Fermi resonances are captured accurately. In our case, the ONIOM QM/MM model was inadequate, unless PCM solvation was also included. This can also be understood directly from the scaling factors: models that reproduced the relative intensities of the two bands (Figure 1 bottom and Figure 4F) had a different scaling factor (0.987–0.988) compared to models that did not reproduce the correct relative intensities (scaling factor 0.975).

## 3. Materials and Methods

MD simulations were carried out using the AMBER 20 software package [57,58]. Lumiflavin parameters were obtained using GAFF [59] and the Antechamber package [60], which is part of AmberTools. Lumiflavin’s charges were determined using the AM1-BCC method [61]. Those charges and GAFF parameters are only relevant to the initial molecular dynamics since lumiflavin’s structure is later refined at the quantum chemical level of theory. Lumiflavin was solvated in a cubic water box described using the TIP3P force field [62]. The cubic solvent box size was selected such that there was at least 3 Å from any atom of lumiflavin to the edge of the box. The effect of changing this solvent shell size from 3 Å to 12 Å was tested, and it was found that changing the box size did not greatly alter the results. Therefore, calculations discussed in this work were carried out for the 3 Å solvent box model unless specified otherwise. The distance cutoff for electrostatics during the simulations was set at 6 Å for the 3 Å solvent shell and 10 Å for the 12 Å solvent shell.

The solvated system was geometry-optimized using the molecular mechanical force field in two steps; first, only the solvent molecules were optimized, keeping lumiflavin fixed, and then the entire system was optimized. The minimized system was used as a starting point for MD simulations. The system was thermalized to 300 K in the canonical (NVT) ensemble over 2 ns. Then, a 5 ns equilibration was carried out with the isothermal–isobaric (NPT) ensemble at a standard pressure of 1 bar. This was followed by a longer 40 ns NPT simulation used to determine the average volume for the simulation. Finally, a production simulation was carried out for 500 ns using the NVT ensemble. The temperature was kept at 300 K, and the volume was kept at the average determined in the NPT step. The distance cutoff for interactions was reduced by 1 Å during the production simulation stage to speed up the calculations.

Snapshots from the MD simulation served as starting points for QM/MM calculations carried out in Gaussian 16 [63] using ONIOM [49]. Electrostatic embedding was used. Since no periodic boundary conditions were used in the QM calculation, all water molecules were initially kept frozen in their MM positions for each snapshot. However, in each snapshot, flavin was optimized at the DFT level of theory using the B3LYP functional and 6-31 + G** basis set. Generally, the B3LYP hybrid functional theory is a popular, well-tested, and cost-effective method for the calculation of ground-state frequencies when scaled using a constant factor [64,65,66,67,68]. The size of the basis set was tested in a previous study on lumiflavin, and B3LYP/6-31 + G** was found to be reasonable for this system, with limited benefit to using a larger basis set [34]. 

To reduce the computational cost associated with optimizing hundreds of QM/MM geometries, optimizations were terminated after a maximum of 10 optimization steps. The maximum RMS gradient for all calculations reported in this work is 0.00702 atomic units, compared to Gaussian’s default threshold of 0.000450. Vibrational frequencies were computed at the same level of theory after hydrogen atoms were replaced with deuterium for all water molecules and for the exchangeable proton on flavin’s N_3_. Computed spectra were adjusted using a constant scaling factor that was chosen for each model to match the computed ν_C=N(out)_ vibrational frequency with the experimental one (1548 cm^−1^). The scaling factor used for each model is indicated in each of the figures in Section 2.

For each computational protocol, 100 snapshots were selected from the last 50 ns of the simulation (at 500 ps intervals). Each of those 100 snapshots was used as a starting point for optimization and frequency calculations with QM/MM. This was done using several different protocols that are represented in Figure 2 and the associated text.

## 4. Conclusions

To more accurately simulate the main features of an IR spectrum of flavin, including broadening and the correct relative frequencies and intensities, a series of increasingly sophisticated molecular models and computational methods were employed. 

In QM/MM calculations for lumiflavin in water, we find that calculations using a single flavin moiety cannot properly represent the ensemble of molecules and solvent configurations around the molecule. Through sampling, we simulated the inhomogeneous broadening of the spectra and provided insights into the intensity changes of specific vibrational modes of lumiflavin due to solvation. Reproducing the relative intensities of bands requires a method and solvation model that correctly reproduce the absolute differences in frequencies of both intense and non-intense bands in the IR spectrum such that Fermi resonances are captured correctly. In our models, a simple point charge description of the solvent is inadequate. We found that for lumiflavin in solution, using a few quantum mechanically treated water molecules nearby as well as a hybrid ONIOM/PCM approach to treat long-range electrostatic effects of the solvent is necessary to simulate the experimental FTIR spectrum. We note also the importance of long-range interactions for simulating other properties in the condensed phase, such as UV–visible spectra and redox or ionization potentials [35,69,70,71,72,73,74,75].

The next step would be to test the computational protocols developed in this study to simulate the FTIR spectrum of flavin embedded in a protein cavity. We expect, however, that due to the lower dielectric environment of a protein, PCM may no longer be necessary to properly capture the relative intensities of IR bands; only a few vibrational frequencies will be affected by specific interactions (e.g., hydrogen bonding) with the protein, compared to a solvent where many frequencies are shifted by a strong dielectric environment. Indeed, ONIOM QM/MM calculations have successfully been shown to reproduce both the relative vibrational frequencies and intensities of quinones in protein binding sites [76].

## Figures and Tables

**Figure 1 molecules-29-00520-f001:**
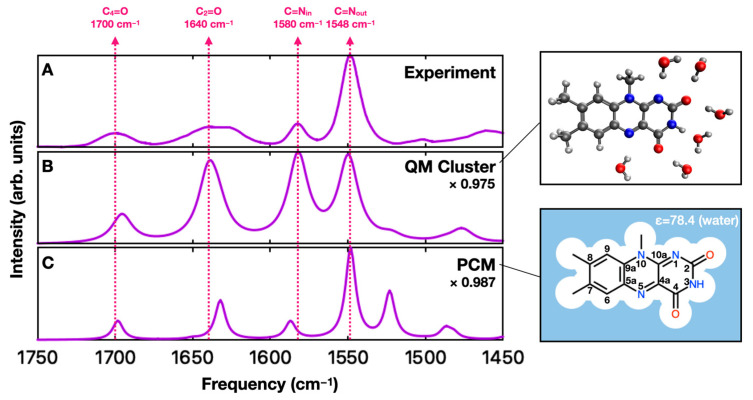
(**A**) Experimental FTIR spectrum for FMN in D_2_O in the 1450–1750 cm^−1^ region from Ref. [24]. (**B**,**C**) Calculated infrared spectra for lumiflavin obtained using a QM cluster (**B**) and PCM (**C**) solvent model from Ref. [34]. The constant scaling factor used to match the experimental C=N_out_ frequency is shown for each calculation. Dashed lines indicate peak positions in the experimental spectra. The insets (**right side**) are schemes of the molecular model used in calculations. Carbon, nitrogen, oxygen, and hydrogen atoms are represented using grey, blue, red, and white, respectively.

**Figure 2 molecules-29-00520-f002:**
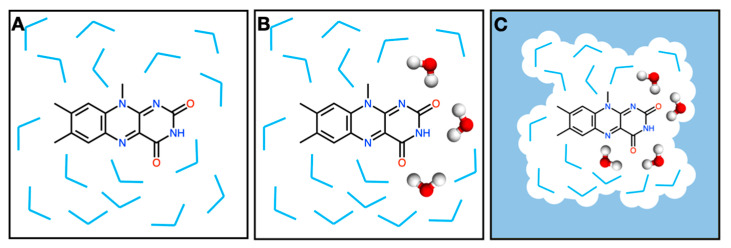
QM/MM protocols used. (**A**) Protocol **M1**: Only the lumiflavin is treated at the QM level of theory. Water molecules are all treated at the MM level and kept frozen in the structure obtained from MD. (**B**) Protocol **M2**: Water molecules that are within 3.5 Å of the lumiflavin carbonyl oxygen atoms are included in the QM region and optimized. The remaining water molecules are treated at the MM level. (**C**) Protocol **M3**: The ONIOM-PCM/X approach is also used to solvate the QM/MM system implicitly. See text for more details.

**Figure 3 molecules-29-00520-f003:**
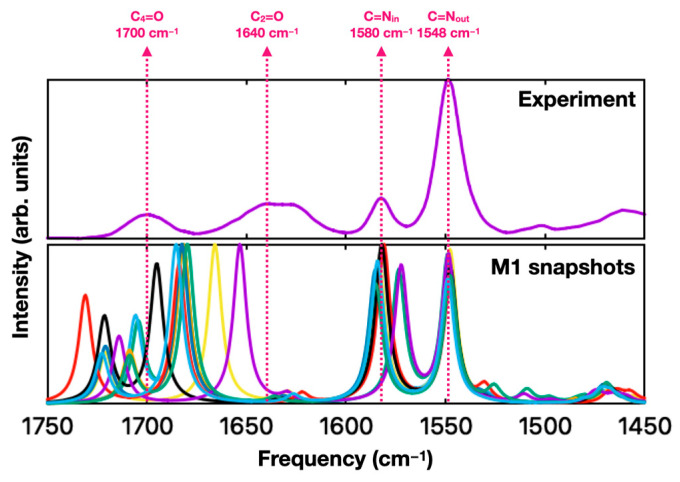
(**Top**) Experimental FTIR spectrum for FMN in D_2_O, in the 1450–1750 cm^−1^ region (reproduced from Figure 1A) [24]. (**Bottom**) Overlay of ten IR spectra computed for lumiflavin using protocol **M1**. The ten spectra, shown using different colors, are obtained starting from different snapshots of an MD simulation. Each spectrum shown here is normalized using a constant scaling factor to match the experimental C=N_out_ frequency. Dashed lines indicate peak positions in the experimental spectra.

**Figure 4 molecules-29-00520-f004:**
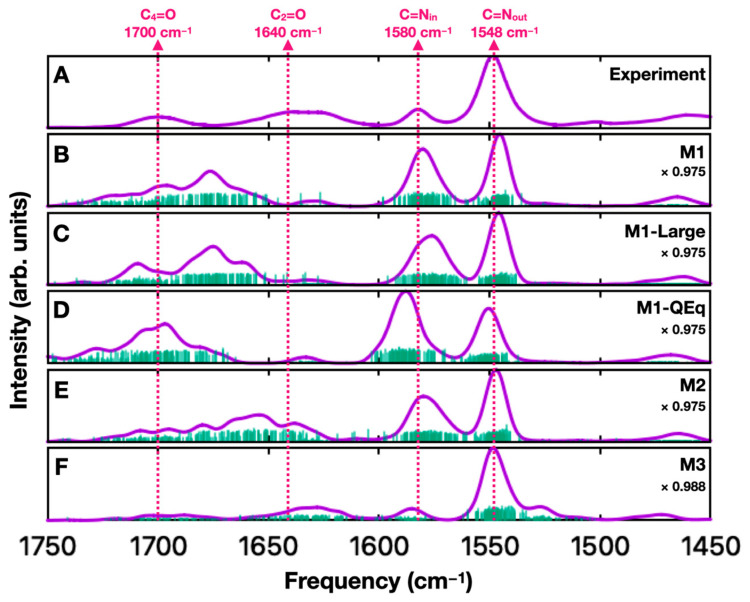
(**A**) Experimental FTIR spectrum for FMN in D_2_O, in the 1450–1750 cm^−1^ region (reproduced from Figure 1A) [24]. Panels (**B–F**) show the combination of 100 FTIR spectra computed for lumiflavin using protocols **M1**, **M1-Large**, **M1-QEq**, **M2**, and **M3**, respectively. The 100 spectra are obtained starting from different snapshots of an MD simulation. Frequencies and intensities from individual calculations are indicated as impulse lines in green. These lines are convolved with an 8 cm^−1^ wide (FWHM) Gaussian function and then summed to give the final computed spectrum (purple). Frequency scaling factors are indicated for each model. Dashed lines indicate peak positions in the experimental spectra.

**Figure 5 molecules-29-00520-f005:**
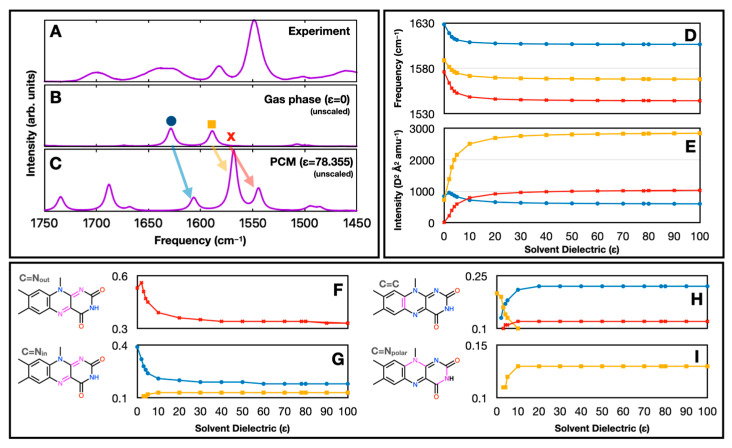
(**A**) Experimental FTIR spectrum of FMN in D_2_O [24]. (**B**,**C**) Computed FTIR spectra for lumiflavin in the gas phase (**B**) and in PCM (**C**). No scaling is used for the calculated frequencies, so the PCM- and gas-phase-calculated spectra can be compared directly. (**D**,**E**) A plot of how the frequencies (**D**) and intensities (**E**) of the three modes labeled as a blue circle, yellow square, and red x in panel (**B**) change as a function of the solvent dielectric constant. (**F**–**I**) VEDA-calculated fractional contributions of specific bond stretching coordinates to the three normal modes, as a function of solvent dielectric. The stretching bonds involved are colored in magenta in the insets.

## Data Availability

Data are contained within the article.

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
