# Peer review of "How Aqueous Solvation Impacts the Frequencies and Intensities of Infrared Absorption Bands in Flavin: The Quest for a Suitable Solvent Model"

_molecules, 2024, doi:10.3390/molecules29020520_

Round 1

Reviewer 1 Report

Comments and Suggestions for Authors

The manuscript describes details of theoretical modeling of the IR spectrum of lumiflavin. The basic structure of this molecule is found in several biologically important systems. The focus of the study is on the test of several theoretical procedures to reproduce the characteristic features in the IR spectrum of lumiflavin in aqueous solution. The authors find the to achieve good accord between experimental relative infrared frequencies and intensities with theoretical predictions requires the application of a sophisticated procedure to account for effects of environment on the computed spectra. The methodology involves (procedure 3 in the authors’ study) several steps. Initial structure is obtained from MD simulations. This is followed by the application of B3LYP/6-31+G** QM/MM ONIOM/PCM computations using the TIP3P model for the surrounding water molecules. Altogether, the conducted research provides insights into the complexity of approaches needed to obtain satisfactory theoretical prediction of IR frequencies and intensities in solution. While the treatment of vibrational frequencies is well established, the predictions of IR intensities in solution is a fundamentally difficult task.

It has been previously shown that satisfactory predictions of IR intensities in the gas phase by using DFT computations is strongly dependent on the both DFT method and basis set quality (DOI: 10.1039/b810877c, DOI: 10.1021/jp108057p). It would be useful if the authors discuss the selected quantum mechanical approach in their investigation in the light of the of the results reported in the above-mentioned studies.

The expression in line 86 “Using our own…”.. is not appropriate.

Reviewer 2 Report

Comments and Suggestions for Authors

In this work different computational methods have been employed to reproduce the FTIR spectrum of lumiflavin, which represents a suitable model for more complex biomolecules. In particular, the FTIR spectrum of the flavin mononucleotide (FMN) in deuterated water has been considered as the experimental reference. The authors clearly demonstrated that to reasonably reproduce the FTIR spectrum - both band positions and relative intensities - a combined used of quantum mechanical/molecular mechanical (QM/MM) and polarizable continuum models (PCM) is needed. The methods employed are clearly described and the main conclusions are supported by data analysis and interpretation. In my opinion, the study clarifies interesting aspects connected to the simulation of FTIR spectra, which might be relevant for the study of biomolecules in solutions. Moreover, the study can stimulate further investigations on related systems. In summary, I recommend the manuscript for publication.

I only have minor comments/suggestions:

The authors emphasized the relevance of long-range electrostatic effects to correctly simulate the FTIR spectrum, but an explanation of this effect has been only partially suggested based on the results depicted in Fig. 5.

In this respect, the role of water molecules H-bonded to specific sites, which must be very relevant, has been not emphasized enough. I wonder if and to what extent, including the polarizable continuum might affect solute-water interactions that, in turn, could modulate relevant spectral features.

In connection with the previous comment, it could be interesting to specify the number of water molecules considered in the QM calculations (M2 and M3 protocols) and, if possible, to provide some structural information on how these water molecules are linked to the solute. Due to the nice agreement between experiment and calculation this might represent an interesting information for those who are interested in solvation (hydration) properties of biomolecules.

Very minor corrections:

Line 27: “in the condensed phase (e.g., in solution or in proteins)”. I would remove “or in proteins” that is not an obvious exemplification of a “condensed phase”.

Line 30: “ condensed-phase” should be “condensed phase”.
